



# Hydrological control of dissolved organic carbon dynamics in a rehabilitated *Sphagnum*–dominated peatland: a water-table based modelling approach

Léonard Bernard-Jannin[1,2,3], Stéphane Binet[1,2,3,4], Sébastien Gogo[1,2,3], Fabien Leroy[1,2,3], Christian
Défarge[1,2,3,5], Nevila Jozja[5], Renata Zocatelli[5], Laurent Perdereau[1,2,3], Fatima Laggoun-Défarge[1,2,3]

[1]Université d'Orléans, ISTO, UMR 7327, 45071, Orléans, France
[2]CNRS, ISTO, UMR 7327, 45071, Orléans, France
[3]BRGM, ISTO, UMR 7327, 45071, Orléans, France
[4]ECOLAB, Université de Toulouse, CNRS, UPS, INPT – UMR 5245, Toulouse, France
[5]CETRAHE, Université d'Orléans, 45072, Orléans, France

*Correspondence to*: Léonard Bernard-Jannin (l.bernardjannin@gmail.com)

## Abstract

Hydrological disturbances could increase dissolved organic carbon (DOC) exports through runoff and leaching, reducing the potential carbon sink function of peatlands. The objective of this study was to assess the impact of hydrological restoration on hydrological processes and DOC dynamics in a rehabilitated *Sphagnum*–dominated peatland. A conceptual hydrological model calibrated on the water table and coupled with a biogeochemical module was applied to La Guette peatland (France), which experienced a rewetting action on February 2014. The model (ten calibrated parameters) reproduced water table and pore water DOC concentration time series (01/04/2014 to 15/07/2017) in two contrasted locations (rewetted and control) of the peatland. Hydrological restoration was found to impact the water balance through a decrease in slow deep drainage and an increase in fast superficial runoff. Observed DOC concentrations were higher in summer in the rewetted location compared to the control and were linked with a difference in dissolved organic matter composition analyzed by fluorescence. Hydrological conditions, especially the severity of the water table drawdown, were identified as the major factors controlling DOC concentration dynamics. The results of the simulation suggest that the hydrological restoration did not affect DOC loads, at least on a short-term period (3 years). However, it impacted the temporal dynamics of DOC exports, which were the most episodic and mainly transported through fast surface runoff in the area affected by the restoration while slow deep drainage dominated DOC exports in the control area. In relation with dominant hydrological processes, exported DOC is expected to be derived from more recent organic matter of the top peat layer in the rewetted area than in the control area. Since it is calibrated on water table and DOC concentration, the model presented in this study proved to be a relevant tool to identify the main hydrological processes and factors controlling DOC dynamics in different areas of the same peatland. It is also a suitable alternative to a discharge calibrated catchment model when the outlet is not easily identifiable.



# 1 Introduction

*Sphagnum*–dominated peatlands represent a major stock of the global soil carbon (C) pool (Gorham, 1991). Dissolved organic carbon (DOC) exports through runoff and leaching could account for up to 25% of the C fluxes (Yu, 2012), reducing the potential C storage function of peatlands (Billett et al., 2004) and impacting downstream water quality (Ritson et al.,

2014). DOC dynamics in peatlands has been found to be strongly controlled by site hydrology, especially by the water table depth (WTD) (e.g. Hribljan et al., 2014; Jager et al., 2009; Strack et al., 2008, 2015). Therefore, hydrological disturbances such as drainage can lead to increased DOC exports in relation with WTD variations (Strack et al., 2008; Worrall et al., 2007). Where disturbances have occurred, hydrological restoration can be undertaken to reestablish peatland functioning (Menberu et al., 2016), with a potential impact on DOC dynamics and exports (Glatzel et al., 2003; Strack et al., 2015;

Worrall et al., 2007).

In peatlands, as in many terrestrial ecosystems, DOC dynamics is controlled on the one hand, by its production to consumption ratio in pore water, and, on the other hand, by lateral water fluxes that drive its exports. DOC production through organic matter decomposition is known to increase with temperature (Clark et al., 2009; Freeman et al., 2001) and DOC consumption, mainly due to heterotrophic bacterial activity, is also positively correlated to temperature and can lead to

15 decreased DOC concentrations during drought (Clark et al., 2009; Pastor et al., 2003). The export of the DOC produced in pore water is mainly controlled by peatland hydrology (Pastor et al., 2003; Strack et al., 2008), especially by the partitioning between quick near surface flow and groundwater flow (Birkel et al., 2014).

While changes in DOC net production resulting from WTD drawdown can be assessed through field monitoring, the relative contributions of DOC production and consumption cannot be evaluated (Strack et al., 2008). Process-based biogeochemical

models can be relevant tools to understand DOC dynamics (Evans et al., 2005) and can help to identify factors controlling its production and consumption in such environments. In particular, conceptual models are appropriate because they are parsimonious in terms of their number of calibrated parameters, avoiding overparametrization issues (Birkel et al., 2017; Seibert et al., 2009). Another advantage of using conceptual models is that they usually require common measured data (e.g. precipitation and water discharge or water level) so they can be applied to numerous study sites where such data are

available, making them a suitable tool to compare sites with different settings.

When studying DOC dynamics in peatlands, existing conceptual models are composed of a DOC module combined with a hydrological model (Birkel et al., 2014; Futter et al., 2007; Lessels et al., 2015). In these studies, the hydrological model is usually adapted to the catchment and calibrated on stream discharge. However, stream discharge in peatlands is difficult to monitor because the diffuse runoff that occurs in these flat areas can result in multiple outlets. Furthermore, while WTD is a

30 key parameter to explain DOC dynamics (Strack et al., 2008), it is usually not considered for calibration, and water discharge is preferred instead. Therefore, while these models have proven to be well adapted when modelling a catchment containing a peatland area (Birkel et al., 2014; Futter et al., 2007; Lessels et al., 2015), where the outlet is well defined, they are more difficult to apply when considering the peatland alone. In this case, the model should focus on the simulation of the



WTD, especially when studying DOC dynamics in peatland pore water. Furthermore, a model based on WTD can also provide interesting information about the spatial variability of the dominant hydrological processes when applied to different locations within the same peatland. Models simulating DOC dynamics are usually based on a simple mass balance and DOC production and consumption rates, usually expressed as first order rate processes (Birkel et al., 2014; Futter et al., 2007; Lessels et al., 2015). In these cases, DOC production and consumption equations are modified using terms related to temperature and soil moisture.

In this study, we propose to couple an existing WTD dependent hydrological model specially developed for simulating peatland hydrology (Binet et al., 2013) with a biogeochemical module simulating DOC production and consumption as first order rate processes. The hydrological model was calibrated on WTD which is an important driver of the DOC dynamics in peatland. The model was applied to two sites of a *Sphagnum*-dominated peatland, one of them having experienced a rewetting action. The objectives were to identify the main hydrological processes and the factors controlling DOC dynamics in the study sites and to assess the impact of the rewetting on DOC export in a *Sphagnum*-dominated peatland.

## 2 Material and methods

### 2.1 Study area and data collection

#### 2.1.1 Site description

The La Guette peatland (150 m.a.s.l., 47°19N, 2°16'E, 20 ha), located in the Sologne forest (Neuvy-sur-Barangeon, France) is an acidic fen mainly composed of moss patches (*Sphagnum cuspidatum*, *S. rubellum* and *S. palustre*) and of *Calluna vulgaris* and *Erica tetralix*. The peatland has been invaded by *Molinia caerulea* and *Betula spp* for 70 years with an acceleration of the invasion in recent decades (Gogo et al., 2011). This was partly caused by a road ditch located near the outlet that accelerated the peatland drainage (Fig. 1). In February 2014, hydrological restoration was undertaken in the road ditch to raise the WTD and reduce its fluctuations in order to promote soil rewetting.

#### 2.1.2 Data collection and analysis

WTD and DOC concentrations ([DOC]) in pore-water were monitored in two locations of the peatland. One is affected by the restoration work and is called "rewetted" while the other is not and is called "control" (Fig. 1). WTD were recorded in piezometers since February 2014 at a 15 min time step using vented-pressure probes (Orpheus mini, OTT Hydromet). Pore-water was sampled in 4 wells surrounding each piezometer (each of them less than 5m from the piezometer) during 12 campaigns that took place every 1 to 4 months between February 2014 and July 2017. The water samples were filtered using 0.45 μm PES filters on the field and transported in an ice box to the lab where DOC concentrations were determined with a TOC analyzer (TOC-L, Shimadzu) within 2 days following sampling (samples stored at 4°C).



Pore water dissolved organic matter (DOM) was characterized by its fluorescence properties through three-dimensional excitation emission matrices (EEMs; Fellman et al., 2010) acquired with F-2500 and F-7000 spectrofluorometers (Hitachi). EEMs were recorded using a 10 x 10 mm quartz mirrored cell, at a photomultiplier voltage of 400 V, with a scan speed of 1500 nm/min, over ranges of excitation of 220–500 nm, in 10 nm steps, and emission of 230–550 nm, in 1 nm steps,

respectively; the slit widths of both monochromators were set at 5 nm. A parallel factor analysis (PARAFAC) was performed using the drEEM toolbox according to the processing described in Murphy et al. (2013). The method was applied to analyze the samples of two campaigns, those of March 2015 (wet conditions) and September 2015 (dry conditions) in order to compare DOM composition for two contrasted hydrological settings.

Meteorological data were recorded at an hourly time step from a station located within the peatland between the two studied

areas (Fig. 1). Rainfall was measured with a tipping bucket rain gauge and potential evapotranspiration (PET) computed with the FAO Penman-Monteith equation at an hourly time step (Allen et al., 1998) using local solar radiation, wind speed, relative humidity and temperature measurements.

The effect of hydrological conditions (dry period from 1$^{st}$ of June to 30$^{th}$ of November and wet period from 1$^{st}$ of December to 31$^{st}$ of May) and location (rewetted or control) on [DOC] and DOM composition were tested using two-way ANOVA and

Tukey's *post hoc* tests were used to identify the locations of the significant differences identified between the factors.

### 2.2 Model description

The modeling approach used in this study combines a conceptual hydrological model with a biogeochemical model simulating DOC dynamics. The hydrological model is based on a conceptual water table dependent hydrological model that has already been successfully applied in the study area (Binet et al., 2013). This model is coupled with a module based on

functions describing DOC production and consumption in pore-water which was developed for this study. The model is described in detail in the following sub sections.

### 2.2.1 Hydrological model

The hydrological model is based on the model described by Binet et al. (2013). It is a daily time step, reservoir model specifically developed for peatland hydrology integrating a WTD dependent runoff. Compared to the original model, a few

modifications were made in this study in order to improve the model. The overall structure of the new model is presented in Fig. 2.

The relation between soil water content and WTD was improved. In the original version the user had to know the relation between WTD and soil water content. Now the model automatically computes the soil water content based on the porosity of the percolation reservoir (Ɵmin), the porosity at the surface (Ɵmax), and peat depth (Hmax) (Fig. 2). The porosity of the

percolation reservoir is considered to be constant over the depth and equal to Ɵmin. The porosity of the Sm reservoir is equal to 0 at the maximum depth (Hmax) and increases linearly with the storage until the surface where it reaches Ɵmax-Ɵmin,





Ɵmax being the total porosity of the Se and Sm reservoirs at the surface. The new relation between WTD and soil moisture content is given by

$$H(\theta) = \frac{\ln(\theta/\theta min)}{\left(\frac{\theta max - \theta min}{Smax}\right)} \qquad (1)$$

$$Smax = \frac{Hmax \, (\theta max - \theta min)}{\ln(\theta max/\theta min)} \qquad (2)$$

where H is the WTD (m) and Ɵ is the sum of the porosities in Sm and Se at a given H.

With this modification, the maximum amount of water stored in the Se reservoir (Semax) that was a calibrated parameter in the original version of the model is now automatically computed with

$$Semax = \theta \min Hmax \qquad (3)$$

Overall, this definition improved the relation between WTD and the water content. In the original version of the model, the porosity of the Sm reservoir was equal to 1, while it now depends on the WTD, to better represent reality (Bourgault et al., 2017).

A third reservoir was added, Sr (overland flow storage), in order to differentiate the overland flow water (Sr) from the water in the peat macroporosity (Sm), that were not differentiated in the original model. While it might not significantly affect the hydrological model, this was done to prepare for the addition of biogeochemical processes which are different for these two reservoirs. Following the addition of this Sr reservoir, a maximum amount of water contained in the Sm reservoir is defined (Smmax) and is computed according to

$$Smmax = Smax - Semax \qquad (4)$$

The routing was also slightly modified to take into account the addition of the new reservoir (Sr). Water from precipitation first fills the Sm reservoir, and the Sr reservoir starts to be filled only when Sm is full (Sm=Smmax). The order in which evapotranspiration is removed from the 3 reservoirs is now Sr, Sm and Se.

Finally a discharge coefficient was added to compute the flow from the new Sr reservoir,

$$O = \alpha o \, Sr \qquad (5)$$

where O is the overland flow from the Sr reservoir (mm), αo is the discharge coefficient of the Sr reservoir (-) and Sr is the volume of water in the Sr reservoir (mm).

This flux is added to the total discharge which is now computed according to

$$Q = D + R + O \qquad (6)$$

where Q is the total discharge (mm), D is the percolation rate from the Se reservoir (mm) and R is the runoff rate from the Sm reservoir (mm).

Given the structure of the model, D represents the drainage of the retention reservoir and can be assimilated to slow deep drainage. R and O represent the drainage of the macroporosity and the overland flow and can be assimilated to fast superficial drainage.

Concerning evapotranspiration, the crop coefficient used to compute evapotranspiration (ET) from ETP was separated into the dormant (Kcd) and the growing (Kcg) season. The latter runs from May to September with a linear relation between the



two coefficients during April and October. This was done to take into account the impact of vascular vegetation growth in peatlands. Finally, a condition was added so that the water level in Sm cannot be lower than the water level in Se.

The computation of the following processes remained unchanged: infiltration from Sm to Se (ISe), percolation (P) and runoff (R). The reader is referred to Binet et al. (2013) for a more detailed description of the computation of these processes.

5    The modified hydrological model is now controlled by 9 parameters (Tab. 1). Three input parameters describing the peat structure (Hmax, Θmin and Θmax) and 6 calibrated parameters controlling water fluxes in the model: Kcd and Kcg for ET, Imax for the ISe, and a discharge coefficient for each reservoir (αp, αr and αo). The forcing variables remained daily precipitation and PET as in the original model.

### 2.2.2 DOC model

To simulate DOC dynamics, a module was developed based on first order production and loss, and mass balance, similarly to what can be found in the literature (Birkel et al., 2014; Lessels et al., 2015). Production and loss are computed in the Se and Sm reservoirs only since the main biogeochemical processes linked to DOC dynamics occur in soil storage and no reaction takes place in the Sr reservoir. DOC production was based on a production coefficient and two additional modifiers based on soil water content and air temperature as usually considered in DOC production models (Birkel et al., 2014; Futter et al., 2007; Lessels et al., 2015). The effect of the temperature was based on a $Q_{10}$ formulation (the factor by which the rate of a reaction increases for every 10-degree rise in the temperature) with a value of 2 according to the value commonly used in DOC production models (Lessels et al., 2015; Michalzik et al., 2003; Tjoelker et al., 2001). The rate modifier based on water content was expressed with a quadratic function to represent the non-linear production of DOC with the variation in soil moisture. Therefore, the higher the soil moisture, the more DOC is produced (Birkel et al., 2014). DOC production is computed as follows:

$$PDOC = k_{prod} \, S \, SOC \, 2^{T/10} \, (S \, / \, Smax \,)^2 \tag{7}$$

where PDOC is the DOC production rate (mg day$^{-1}$ m$^{-2}$), $k_{prod}$ is the production constant (day$^{-1}$), SOC is the amount of organic carbon per mm of peat per square meter (mg mm$^{-1}$ m$^{-2}$), T is the air temperature (°C), S is the amount of water in the considered reservoir (mm) and Smax is the maximum amount of water in the considered reservoir (mm).

DOC loss was based on a loss coefficient and is linked to air temperature in the same way as DOC production. DOC loss is computed according to

$$LDOC = k_{loss} \, [DOC] \, S \, 2^{T/10} \tag{8}$$

where LDOC is the DOC loss rate (mg day$^{-1}$ m$^{-2}$), $k_{loss}$ is the loss constant (day$^{-1}$), [DOC] is the DOC concentration in pore water (mg L$^{-1}$) and S is the amount of water in the considered reservoir (mm).

Finally, the mass balance of DOC is computed in the Sm and Se reservoirs

$$DOC_{Sm}^{t+1} = DOC_{Sm}^{t} + PDOC_{Sm}^{t+1} - LDOC_{Sm}^{t+1} + [DOC]_{rain} ISm - [DOC]_{Sm}^{t} (ISe + R) \tag{9}$$





$$DOC_{Se}^{t+1} = DOC_{Se}^t + PDOC_{Se}^{t+1} - LDOC_{Se}^{t+1} + [DOC]_{Sm}^t ISe - [DOC]_{Se}^t D \qquad (10)$$

where the exponent represents the time step, the subscript indicates the reservoir considered (Sm or Se), $[DOC]_{rain}$ is the DOC concentration in rain water (mg L$^{-1}$), Ise is the infiltration from Sm to Se (mm) and Ism is the infiltration from Sr to Sm (mm).

The DOC model is controlled by 6 parameters (Tab. 1). Two input parameters (SOC and $[DOC]_{rain}$) and four calibrated parameters controlling DOC dynamics (production and loss constant in two reservoirs). The additional forcing variable is air temperature.

### 2.2.3 Model setup

The hydrological and biogeochemical model parameters were calibrated for each piezometer of the peatland for the period ranging from 01/04/2014 to 01/05/2016. This period was chosen because it includes a relatively wet (2014) and relatively dry (2015) summer. The validation period started from 01/10/2016 to 15/07/2017. The period from 01/05/2016 to 30/09/2016 was not simulated because exceptionally heavy rainfall occurred on 31/05/2016, causing extensive flooding in the whole region. The definition of the model is not suitable for these exceptional events because the water coming from flooded rivers is not taken into account in the model. Өmin and Өmax were set at 0.2 and 1, respectively, and Hmax at 0.6 m, based on field data. $[DOC]_{rain}$ was 2 mg L$^{-1}$ according to measurements performed on rain water and SOC was set at 833 10$^3$ mg mm$^{-1}$ m$^{-2}$ following measurements performed on peat samples.

### 2.2.4 Model evaluation

The parameters were calibrated with a Nelder-Mead algorithm (Varadhan et al., 2016) implemented in the R software (R Core Team, 2012) using the Nash-Sutcliffe coefficient (NS, Nash and Sutcliffe, 1970) as the objective function for the hydrological module and the root-mean-square error (RMSE) for the DOC concentrations. The hydrological module was calibrated first because substantially more water table data were available than DOC concentrations. The DOC module was then calibrated over the calibrated hydrological model. Sensitivity analysis was performed using a latin-hypercube one-factor-at-a-time (LHOAT) procedure (Zambrano-Bigiarini and Rojas, 2014) implemented in the R software. The sensitivity analysis was based on NS for the hydrological model and on RMSE for the DOC model.

### 3 Results

#### 3.1 Observed hydrology and DOC

The mean annual precipitation (P) of the area was 821 mm yr$^{-1}$ and the mean annual PET 931 mm yr$^{-1}$ for the period ranging from 01/04/2014 to 01/04/2016 (Tab. 2). WTD and DOC exhibited different dynamics between rewetted and control areas (Fig. 3b, c). The water table was close to the surface level in each piezometer during the wet season but the length of this





season depended on the severity of the water table drawdown that occurred during the previous drier season. In 2014, a particularly wet year (P=906 mm and PET=904 mm from 01/04/2014 to 01/04/2015), the water table reached the surface in December 2014 while for the following season, which was relatively dry (P=736 mm and PET=960 mm from 01/04/2015 to 01/06/2016), it reached the surface in May 2016. The WTD was lower on average and with a greater variability in the control than in the rewetted area but the main difference between the sites was the severity of the maximum water table drawdown which was 19 cm in the rewetted and 43 cm in the control site, with the same climatic conditions for both locations.

The average of [DOC] measurements was $13.3\pm4.6$ mg L$^{-1}$ in the control site and $21.6\pm7.2$ mg L$^{-1}$ in the rewetted one. [DOC] were globally higher in the rewetted than in the control site (p<0.001) but this was especially true in the dry period. Overall, [DOC] were higher in dry periods than in wet periods for the rewetted site while this difference was not observed in the control site (Fig. 4a). Finally, when considering the temporal evolution of [DOC], the main difference was observed between April and October 2015 where [DOC] rose in the rewetted but decreased in the control site (Fig. 5).

The PARAFAC analysis revealed three main components characterizing the DOM (Fig. 4b). According to the review by Fellman et al. (2010), the first component (ex 360, em 466) can be described as high-molecular-weight and humic and is referred to here under its original name as C. The second component (ex 330, em 407) can be described as low-molecular-weight and is referred to here as M. The third component (ex 250, em 446) can be described as high molecular weight and humic and is referred to here as A. Component A is known to be more aromatic than C (Fellman et al., 2010), even if in our case, the shorter emission wavelength for component A than for C may also indicate that C is more aromatic than A (McKnight et al., 2001). The ratio of the contribution of component C over the contribution of A and of the contribution of component M over the contribution of A in pore water samples of the wet and dry campaigns are presented in Fig. 4 (c and d). A large increase in the contribution of C relative to the contribution of A was observed in dry conditions in the rewetted area (p<0.001) while the ratio was similar for control and rewetted sites in wet conditions. Similarly a significant increase in the contribution of M relative to the contribution of A was observed during dry conditions in the rewetted site compared to wet conditions in control and rewetted areas (p<0.001).

## 3.2 Hydrological modeling

Simulated and observed WTD dynamics are shown in Fig. 3. Overall the model performed well for both locations with NS greater than 0.3 for all validation and calibration periods and reaching values greater than 0.8 for calibration periods (Tab. 3). The decrease in the model efficiency in validation compared to calibration periods can be explained by the exceptional event that occurred prior to the validation period and that may have disturbed the hydrological balance. The most important point is that the model is able to reproduce two different WTD dynamics using the same input data. These differences are explained by the difference in calibrated parameter values. The evapotranspiration coefficient, maximum infiltration rates and Se discharge coefficient are higher in the control than in the rewetted site, and Sr and Sm discharge coefficients are higher in the rewetted site than in the control one (Tab. 3). These differences are reflected in the water balance of each




location with a higher evapotranspiration and slow deep drainage in the control than in the rewetted location, and a higher fast superficial drainage in the rewetted than in the control one (Tab. 2). Sensitivity analysis indicates that the model is the most sensitive to the evapotranspiration coefficient in the growing season and the Se discharge coefficient, and the least sensitive to the evapotranspiration coefficient in the dormant period and the Sr discharge coefficient for both locations (Tab. 4).

### 3.3 DOC dynamics modeling

Simulated and observed pore water [DOC] are shown in Fig. 5. Overall, the model performed well considering its simplicity (4 calibrated parameters), with RMSE < 6 mg L$^{-1}$ for calibration and validation in both rewetted and control sites, and with no systematic overestimations or underestimations. The model performed better for the control than for the rewetted site but as opposed to the hydrological model simulations, the model performed similarly for the calibration and validation periods. The model was able to reproduce [DOC] dynamics in both locations, especially the rising concentrations in the rewetted site and the decreasing concentrations in the control site during summer 2015 (Fig. 5). The sensitivity analysis indicates that the model was the most sensitive to parameters related to the Se reservoir and the least sensitive to parameters related to the Sm reservoir for both locations (Tab. 5). Compared to the hydrological model, DOC related parameters were not greatly different between rewetted and control areas (Tab. 6). The model can compute the DOC balance for each location which is shown in Tab. 2. Overall, DOC production, loss and exports were similar and in the same order of magnitude for each location. Nevertheless, a difference can be observed for the partitioning between exports from the Se and Sm reservoirs. While all the exports are driven by Se in the control site, exports from Se only account for 36% of the total DOC exported in the rewetted site (Tab. 2). Fig. 6 gives the temporal dynamics of simulated DOC exports for each location, showing that DOC exports are less variable in Se than in Sm where large peaks of DOC exports can be observed.

## 4 Discussion

### 4.1 Hydrological processes

In this study, observed water table dynamics were used to better understand the dominant hydrological processes taking place in two locations of a restored peatland (rewetted and control), by calibrating a conceptual model. Though simple (6 calibrated parameters), the model was able to correctly reproduce the specific water table dynamics in each location of the studied area, using the same input data (precipitation and potential evapotranspiration). This difference in observed water table dynamics (24 cm of difference for the maximum water table drawdown) is reflected in the calibrated parameter values for each location (Tab. 3). In addition, and in order to better assess the dominant processes, a sensitivity analysis of the model was performed for each location (Tab. 4). The results indicate that the most sensitive parameters are Kcg and αp which are related to the evapotranspiration during the growing season and the deep drainage of the retention reservoir (Se), respectively, meaning that these processes are the most important ones to explain the peatland hydrology. Both parameters





are the highest for the control location and these differences can explain the dissimilarity in the severity of the water table drawdown observed in the two locations. Kcg is 0.54 and 0.22 for control and rewetted locations, respectively. The value of the coefficient for the control location is consistent with the values reported by Lafleur et al. (2005) for a shrub covered bog in Canada (0.517). However, the value of the parameter for the rewetted location is lower than the range of commonly

observed values which is between 0.3 and 0.8 (Lafleur et al., 2005). This difference between the two locations is particularly important considering that the vegetation in both is similar. In this case, low Kcg values can reflect lateral water redistributions which are  rapid lateral exchanges between inundated and non-inundated regions of the peatland, as suggested by Mclaughlin and Cohen (2014). This is likely linked with the restoration work that created an inundated area in its vicinity. It is also in agreement with the results of Wilson et al. (2010) indicating that the frequency of full saturation of the peat

increases markedly after a drain blocking operation. The discharge coefficient of the retention reservoir indicates the intensity of the slow deep drainage of the peatland. This coefficient is higher in the control part than in the rewetted part and this is reflected in the water balance through the partitioning of the total discharge in the two locations. Deep drainage represents the quasi-totality of the total discharge upstream while it accounts for 18% of the total discharge downstream. Similarly to the difference in evapotranspiration coefficients, this difference in the partitioning between fast superficial and

slow deep discharge can also be related to the restoration work, since the blockage of the drain could have reduced the deep drainage and increased the amount of surface drainage in the rewetted area. Therefore, the model helps to characterize the impact of restoration as seen in the water balance and evapotranspiration and deep drainage coefficients. It enables deep drainage dominated (control) and surface drainage dominated (rewetted) systems to be identified within the same peatland, in relation with hydrological restoration work.

**4.2 DOC dynamics control factors**

### 4.2.1 Model results

A module simulating DOC production and loss was added to the hydrological model in order to better understand DOC dynamics in the two peatland locations. Considering the simplicity of the model structure, it gave satisfactory results, with RMSE always smaller than 6 mg $L^{-1}$. However, the quality of the results is more difficult to assess than for the hydrological

model because few data were available for the calibration and validation steps. Nevertheless, it is noteworthy that the model, based on only 4 calibrated parameters, is able to capture the two different dynamics recorded in each location, i.e. rising [DOC] in the downstream location in summer 2015 and a decreasing [DOC] in the upstream location in the same period.

### 4.2.2 DOC concentrations and control factors

Long-term studies have reported decreasing pore water [DOC] more than 10 years after a restoration operation took place

(Höll et al., 2009; Wallage et al., 2006), while others observed increasing [DOC] after restoration (Hribljan et al., 2014; Strack et al., 2015). Glatzel et al. (2003) observed an increase in pore water [DOC] following a drain blocking operation but



predicted a decrease in [DOC] with time due to a depletion of easily decomposable organic matter in the peat. In this study, the results indicate that, during the three years following a restoration operation, [DOC] were higher in the rewetted than in the control location during the dry period (from 1st of June to 30th of November), while they were similar during the wet period. In addition, the difference in [DOC] dynamics is also reflected in DOM quality inferred from its fluorescence

properties, with a greater increase in low molecular weight compounds (component M) and fewer aromatic high molecular weight compounds (component C) in the rewetted location during the dry season compared to the control area. These findings are in agreement with the studies by Höll et al. (2009), Hribljan et al. (2014) and Strack et al. (2015) who observed that wetter sites would result in a pore water with smaller and fewer aromatic dissolved organic molecules (likely sourced from inputs of fresh litter from growing vegetation) than the sites with a lower water table.

The most sensitive parameters (production and loss rates in Se) of the DOC model do not differ between the control and rewetted areas. There are differences in constant rates related to the Sm reservoir but these parameters are the least sensitive, meaning that they would not greatly impact [DOC]. This means that the model is able to explain the difference in [DOC] between the two locations with no differences in production and/or loss rates, suggesting that other factors control DOC dynamics in the peatland. The main difference in [DOC] is observed during the dry period, when the water table dynamics is

different between the two locations. This would confirm that hydrology, and especially the magnitude of the water table drawdown, might be a major factor controlling [DOC] dynamics in the peatland. Indeed, the higher WTD in the dry period in the rewetted site is related with a higher [DOC] than in the control site where the WTD is lower. A larger proportion of low aromatic DOC is also observed during the same period in the rewetted than in the control site. Therefore, we propose to explain the differences in [DOC] by the difference in water table drawdown in the dry period. When the water table

drawdown is small (high water table), more DOC is produced from the top peat layer containing more recent and easily biodegradable organic matter than when the water table drawdown is more severe (low water table). In addition, anaerobic conditions in the rewetted site would lead to less efficient decomposition of organic matter, increasing the production of water-soluble intermediate metabolites (Kalbitz et al., 2000; Strack et al., 2008). An increase in [DOC] in the rewetted location can also be explained by an increase in the photic zone, potentially supporting algae photosynthate production

enhancing DOC release into the water column, as suggested by Hribljan et al. (2014). However the latter hypothesis is the least probable in our case since no ponding water is observed in summer in the study area. The ability of the model to reproduce pore water [DOC] dynamics can be attributed to its consideration of the water table drawdown which is expressed in the model through the use of soil moisture (based on water level in the Sm and Se reservoirs) as a production rate modifier.

### 4.2.3 DOC exports

The model enables DOC exports to be estimated for each location. The results are in the range reported in the literature (from 4.2 to 18.9 g-C m$^2$ yr$^{-1}$, Birkel et al., 2014, 2017 and Jager et al., 2009). DOC exported from the control site is slightly higher than that from the rewetted one but in the same order of magnitude. However, considering the simplicity of the





model, it is difficult to affirm that this difference is significant and that more DOC is exported from the control than from the rewetted site. Nevertheless, the partitioning between DOC exports from the two production reservoirs is clearly different for each location. According to the water balance, DOC exports are only driven by the deep drainage from the Se reservoir in the control site while the amount of DOC exported through deep drainage and runoff is more balanced in the rewetted site.

This clearly reflects the dominant hydrological processes in each location and can be seen in the temporal variability in DOC exports (Fig. 6). DOC exports are more episodic in the rewetted site, with 67% of the DOC load coming from the Sm reservoir and representing only 22% of the total simulated period length. These results are consistent with the results of Birkel et al. (2017) who reported that 60% of the DOC was exported in 30% of the time in a small peat catchment through rapid near-surface runoff. However, in the control site, DOC exports are more constant than in the rewetted one following

the slow but regular deep drainage of the Se reservoir. These results suggest that hydrology has a major impact on DOC load dynamics, since it is the partitioning between superficial quick flow and slow deep drainage that controls the temporal dynamics of DOC exports. This hydrological control on DOC fluxes also affects the source of DOC exported from the peatland, in relation with the difference in DOM composition observed with the fluorescence analysis. Therefore, in the rewetted area the DOC exported will exhibit characteristics of top peat layer recent organic matter (less aromatic) while it is

likely derived from older and deeper organic matter (more aromatic) in the control area. These findings indicate that, while its impact on DOC loads can be negligible, restoration work might have an impact on stream quality by releasing a great amount of DOC during rainfall events. However, this is valid for a three-year period following the restoration and might be different for the future, underlining the need for long term monitoring to correctly assess the impact of hydrological restoration on DOC dynamics.

### 4.3 Perspectives for application of the model

The model developed in this study follows a parsimonious coupled hydrology-biogeochemistry model philosophy (Birkel et al., 2014, 2017; Lessels et al., 2015). By keeping parametrization to a minimum, it was able to identify factors controlling WTD and DOC dynamics in the two contrasted sites of the studied peatland with a relatively low requirement in input data

(precipitation, potential evapotranspiration and temperature). Contrary to similar models, hydrology is here calibrated on WTD instead of on stream discharge. This way, the model proves to be a relevant tool to explore the hydrology of areas located within the same peatland and to highlight the impact of hydrological restoration on hydrology and DOC dynamics that would have been difficult to study with models calibrated on stream discharge and applicable at the catchment scale only. In addition, the DOC model developed in this study has shown good results in modeling pore water [DOC] dynamics,

meaning that the formulation of the 4 calibrated parameters model is adapted to peatland ecosystems. However, as this model is calibrated on pore water [DOC], DOC export results have to be interpreted with care. In order to improve its significance, the model should be compared with a discharge calibrated model on a study site where the outlet is well defined and monitored. Therefore, if applied to several WTD time series, it could provide spatial information by identifying

the main areas of DOC production within a peatland. This model could also be applied to longer time series and different study sites to assess the effect of hydrological restoration over longer periods and the dominant controlling factors in peatlands with different settings.

## 5 Conclusions

A conceptual hydrological model, especially developed for peatland and calibrated on WTD, has been combined with a simple DOC production/loss model and applied to two locations of a peatland, one of them affected by hydrological restoration. The application of this model has shown the following:

- The hydrological restoration was found to impact water balance, by increasing fast superficial drainage compared to slow deep drainage.
- The intensity of the maximum water table drawdown was found to be the main factor controlling pore water [DOC] dynamics in the peatland.
- Higher [DOC] in the rewetted location was linked to differences in DOM composition
- Simulated DOC exports are in the same order of magnitude for rewetted and control locations, in a short-term period (3 years).
- Water partitioning between fast superficial drainage and slow deep drainage controls DOC sources as well as the temporal dynamics of DOC exports

These results suggest that hydrological restoration does not affect short term DOC fluxes in peatland. In addition, this study has shown that the proposed conceptual hydrological and biogeochemical model can provide relevant information about water balance and the factors controlling element cycling processes in peatlands. The application of a WTD based model is a

relevant alternative to a discharge calibrated catchment model when the outlet is not easily identifiable or when seeking for within-peatland spatial information.

*Author contributions*

FLa, SG and SB designed the study site restoration and monitoring.

LBJ, SG, FLa, FLe and LP helped with instrumentation and data collection.

CD, NJ and RZ helped with fluorescence analysis and data interpretation

LBJ and SB developed the model

LBJ performed simulations and data analysis

LBJ prepared the draft of the manuscript

FLe, SG, CD, NJ, FLa and SB helped improve the final manuscript

*Acknowledgments.*



This paper is a contribution of the Labex VOLTAIRE (ANR-10-LABX-100- 01) and of the PIVOTS project (ARD 2020 of the Centre Val de Loire region, CPER and FEDER). This study was undertaken in the framework of the Service National d'Observation Tourbières (French Peatland Observatory), accredited by the INSU/CNRS. The authors would like to thank A. Guirimand-Dufour and F. Le Moing for their help in fluorescence analysis, N. Lottier for DOC analysis and E. Rowley-
Jolivet for revision of the English version.

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





Figures

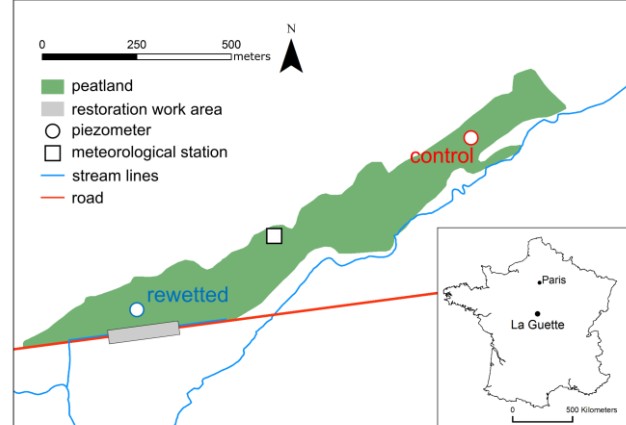

**Figure 1: Location and settings of the study area. Locations of control and rewetted monitoring are indicated.**

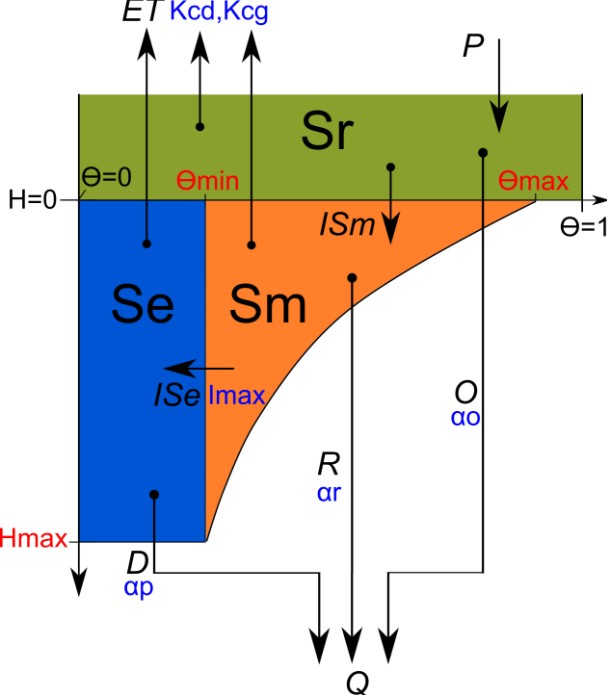

5    **Figure 2: Structure of the hydrological model, composed of three reservoirs, surface (Sr), macroporosity (Sm) and retention (Se). The different fluxes are indicated in italics, P (precipitation), ET (evapotranspiration), ISm (infiltration from Sr to Sm), ISe (infiltration from Sm to Se), D (deep drainage from Se), R (runoff from Sm), O (overland flow from Sr). Total discharge Q corresponds to the sum of D, R and O. Note that given parameters are written in red and calibrated parameters associated to each flux in blue, see description in Tab. 1.**





Figure 3: (a) Time series of meteorological data (PET, potential evapotranspiration and P, precipitation) used as input data in the hydrological model, (b) simulated and observed WTD in the rewetted site and (c) simulated and observed WTD in the control site. Calibration and validation periods are also indicated.



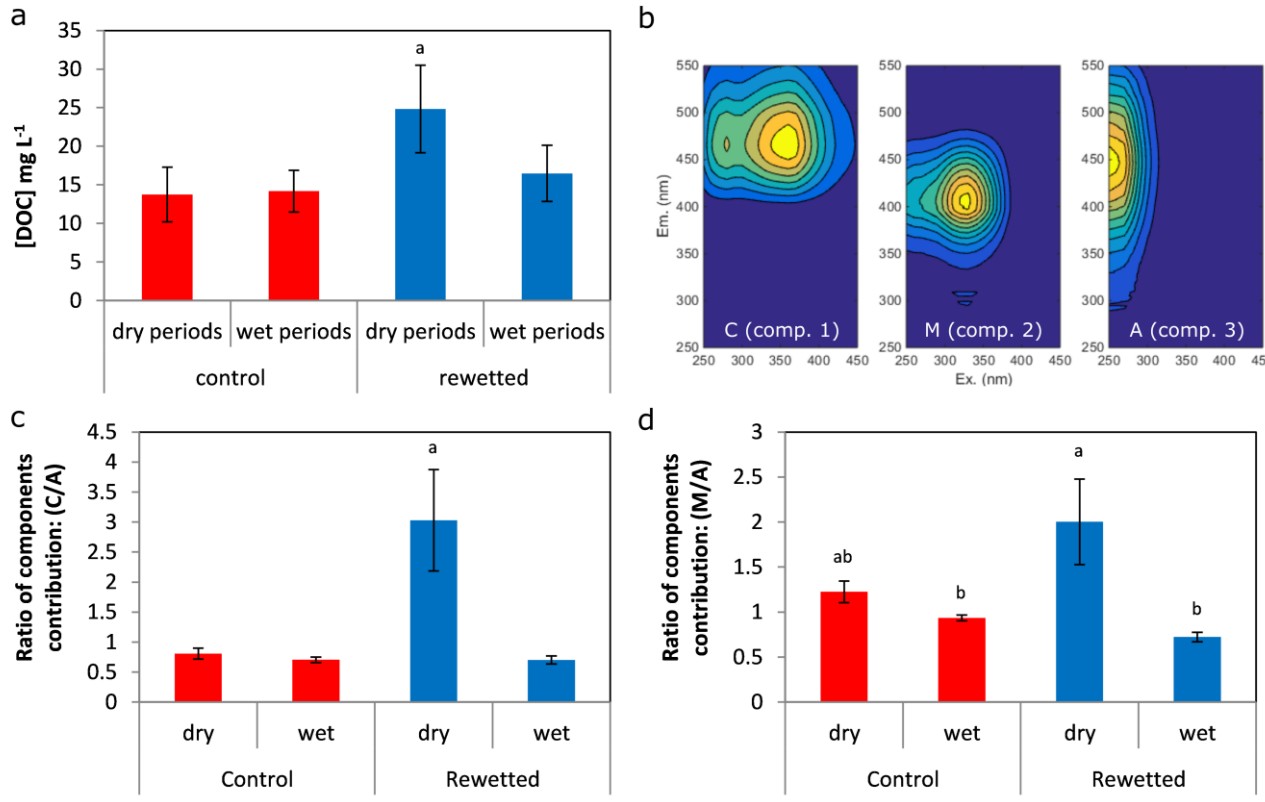

**Figure 4:** (a) DOC concentrations in control and rewetted sites for dry (1$^{st}$ of June to 30$^{th}$ of November, n=7) and wet periods (1$^{st}$ of December to 31$^{st}$ of May, n=6). ). (b) Excitation-emission matrices for the identified PARAFAC components (see the text for details). (c) Ratio of contribution of component C over component A for dry and wet conditions in control and rewetted sites (n=4). (d) Ratio of contribution of component M over component A for dry and wet conditions in control and rewetted sites (n=4). The letter above the bar indicates significant differences across different conditions (Tukey's p<0.01). (b) Excitation-emission matrices for the identified PARAFAC components (see the text for details).

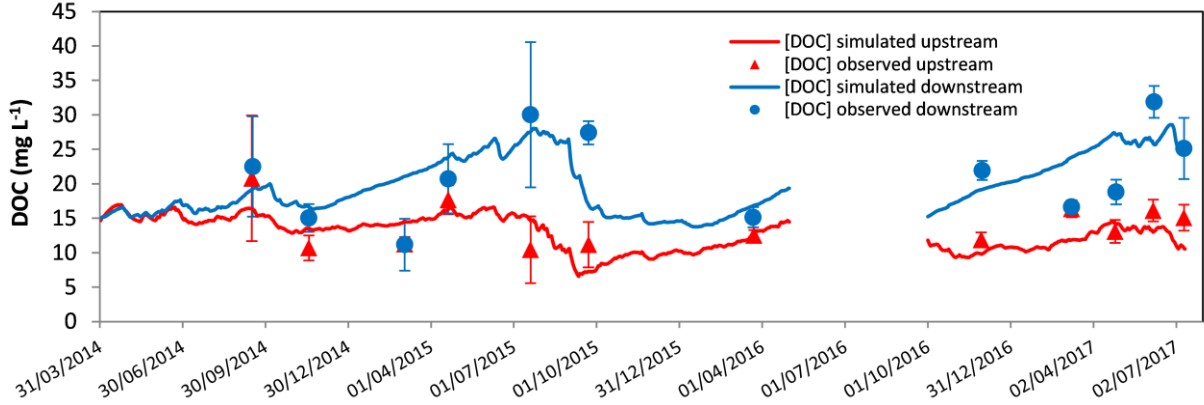

**Figure 5:** Simulated and observed pore water [DOC] in control and rewetted sites. Observations are the average of 4 samples for each sampling date. Error bars indicate standard deviation.





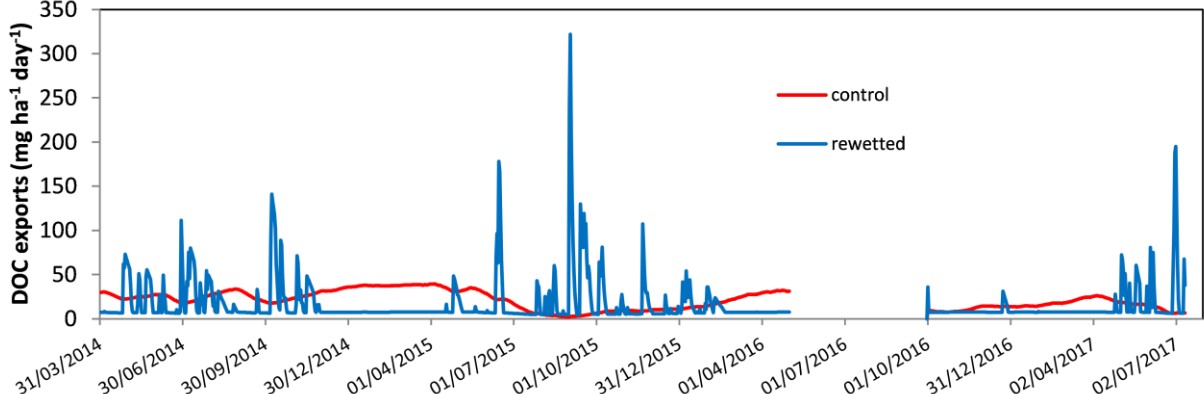

**Figure 6: Simulated DOC exports for control and rewetted sites.**



Tables

**Table 1: List of the parameters used in the hydrological and in the DOC model. The hydrological flux associated to each parameter is in parenthesis. Calibrated parameters are indicated.**

|  | Symbol | Process | Description | Units | Calibrated |
|---|---|---|---|---|---|
| *Hydrological model* | Hmax | WTD-moisture relation | Peat depth | mm | no |
|  | Ɵmin | WTD-moisture relation | Porosity at maximum depth | $m^3.m^{-3}$ | no |
|  | Ɵmax | WTD-moisture relation | Porosity at the surface | $m^3.m^{-3}$ | no |
|  | Kcd | Evapotranspiration (ET) | Crop coefficient for dormant season | - | yes |
|  | Kcg | Evapotranspiration (ET) | Crop coefficient for growing season | - | yes |
|  | Imax | Infiltration Sm to Se (ISe) | Maximum infiltration rates in Se | mm | yes |
|  | αp | Se discharge (D) | Discharge coefficient of Se | $day^{-1}$ | yes |
|  | αr | Sm discharge (R) | Discharge coefficient of Sr | $day^{-1}$ | yes |
|  | αo | Sr discharge (O) | Discharge coefficient of So | $day^{-1}$ | yes |
| *DOC model* | SOC | DOC module | Mass of TOC per height of peat | $mgC\ mm^{-1}$ | no |
|  | DOCrain | DOC module | DOC concentration in rain water | $mg\ L^{-1}$ | no |
|  | kprodSe | DOC module | DOC production coefficient in Se | $day^{-1}$ | yes |
|  | klossSe | DOC module | DOC loss coefficient in Se | $day^{-1}$ | yes |
|  | kprodSm | DOC module | DOC production coefficient in Sm | $day^{-1}$ | yes |
|  | klossSm | DOC module | DOC loss coefficient in Sm | $day^{-1}$ | yes |

5  **Table 2: Water and DOC balance computed from 01/04/2015 to 01/04/2017 in rewetted and control areas. P is precipitation, ET is evapotranspiration, Q is total discharge, O is overland flow, R is macroporosity runoff, D is deep drainage, PDOC is the amount of DOC produced, LDOC is the amount of DOC loss and DOC exports is the amount of DOC exported. Number in brackets represents the DOC balance for Se and Sm reservoirs.**

| 01/04/2015 to 01/04/2017 | rewetted | control |
|---|---|---|
| P (mm yr⁻¹) | 821 | 821 |
| ET (mm yr⁻¹) | 160 | 494 |
| Q (mm yr⁻¹) | 653 | 330 |
| *O (mm yr⁻¹)* | *243* | *1* |
| *R (mm yr⁻¹)* | *295* | *1* |
| *D (mm yr⁻¹)* | *115* | *329* |
| PDOC [Se – Sm] (g-C m⁻² yr⁻¹) | 79.2 [73.7-5.5] | 80.8 [80.4-0.4] |
| LDOC [Se – Sm] (g-C m⁻² yr⁻¹) | 74.0 [74.0-0] | 74.6 [74.6-0] |
| DOC exports [Se – Sm] (g-C m⁻² yr⁻¹) | 6.7 [2.4-4.3] | 8.2 [8.2-0] |





**Table 3: Calibrated parameters and efficiency of the hydrological model**

| Parameter | Units | rewetted | control |
|---|---|---|---|
| Kcd | - | 0.0120 | 0.500 |
| Kcg | - | 0.221 | 0.540 |
| Imax | mm | 0.955 | 2.88 |
| $\alpha p$ | day$^{-1}$ | 1.13E-3 | 4.21E-3 |
| $\alpha r$ | day$^{-1}$ | 0.422 | 0.001 |
| $\alpha o$ | day$^{-1}$ | 0.258 | 0.001 |
| NS calibration | - | 0.80 | 0.86 |
| NS validation | - | 0.33 | 0.57 |

**Table 4: Sensitivity rank of the parameters of the hydrological model**

| Parameter | *Sensitivity rank* | |
|---|---|---|
|  | rewetted | control |
| Kcg | 1 | 1 |
| $\alpha p$ | 2 | 2 |
| $\alpha r$ | 4 | 3 |
| Imax | 3 | 4 |
| $\alpha o$ | 6 | 5 |
| Kcd | 5 | 6 |

**Table 5: Sensitivity rank of the parameters of the DOC model**

| Parameter | *Sensitivity rank* | |
|---|---|---|
|  | rewetted | control |
| kprodSe | 2 | 1 |
| klossSe | 1 | 2 |
| kprodSm | 3 | 3 |
| klossSm | 4 | 4 |



**Table 6: Calibrated parameters and efficiency of the DOC model**

| Parameter | Units | rewetted | control |
|-----------|-------|----------|---------|
| kprodSe | day$^{-1}$ | 1E-6 | 1.7E-6 |
| klossSe | day$^{-1}$ | 3.7E-2 | 3.9E-2 |
| kprodSm | day$^{-1}$ | 7.2E-8 | 1E-8 |
| klossSm | day$^{-1}$ | 1E-6 | 1E-5 |
| RMSE calib | mg L$^{-1}$ | 5.7 | 3.3 |
| RMSE valid | mg L$^{-1}$ | 5.3 | 3.9 |