# Peer review of "Hydrological control of dissolved organic carbon dynamics in a rehabilitated *Sphagnum*-dominated peatland: a water-table based modelling approach"

_Hydrology and Earth System Sciences, 2017_

## Referee Comment (RC1) · Anonymous Referee #1 · 27 Nov 2017

The ms describes a modeling approach to predict the relationship between DOC production and hydrology, especially water table levels, in a fen which partly experienced draining and restoration. The authors used a parsimonious approach including a hydrological and a biogeochemical component. The model was calibrated directly at the site through monitoring of the peatlands water table level at the two measuring sites and by DOC analyses. I addition the authors carried out fluorescence measurements to determine changes in the DOM quality. In principal, several parsimonious models to predict the release of DOC dependent on hydrological changes from peatlands ex-

ist. Different from these models which commonly use discharge data to characterize the hydrological component, the authors use changes in water table to predict DOC release. The authors concluded that their model predicts DOC release at the two sites reasonably well (whatever this means). As an advantage compared to other models they argue that the use of water table changes instead of using discharge measurements allows predicting local changes in DOC release esp. if parts of a peatland underwent draining and restoration, which might have strong influence on DOC production and release. I don't have an emphasis on hydrological modeling of peatlands but on the biogeochemical processes of DOM production on peatlands, so I will not comment on the quality of the model itself. I had a bit a hard time with the structure of the paper. One reason might be that the mentioned objectives of the study are very general .... to identify hydrological processes and factors controlling DOC dynamics ......the impact of rewetting on DOC export. It is known that water table changes is the major driver of DOC release in peatlands, because surface wetness determines the redox conditions on the mire's surface which in turn control microbial activity and peat decomposition. In this sense, it would be helpful if the authors could first describe in more detail what they expect the effect of peatland restoration on DOC release could be, especially what is expected how the amount and quality of DOM changes as result of drainage and subsequent rewetting. Based on this it would be easier to understand the chosen approach. Moreover, I think that the comparison between the WTD and the stream discharge approach should be discussed in more detail. On page 12 the authors argue that the advantage of the WTD approach is that this model can predict DOC release at different areas in the same peatland in contrast to the stream discharge approach which integrates the entire peatland. However, the authors have only selected a single site with is under restoration and did not define which biogeochemical or hydrological component/factor is specific for the rewetted site. Due to this, it remains unclear how the specific behavior of the rewetted site compares to the overall variation of DOC release and DOM quality during water table changes in the entire peatland (compare the discussion in Birkel et al 2017 and Broder et al . 2015 cited therein).

This means that the observed changes at the single rewetted site provide a small data set only, so that it remains somehow speculative if the shown changes and effects are specific for rewetting sites. A larger data set which shows the general variability of WTD, DOM quality and DOC export at the selected peatland compared to the specific rewetted site would convince me that the chosen approach is of advantage compared to the one using the stream discharge approach. I suggest that the authors discuss this in more detail including effects of spatial varying redox conditions (also hummocks and hollows), plant cover (what means Sphagnum dominated in this context), mineral redox barriers esp. iron-oxides; which are important for DOC dynamics in fens (is this site minerothrophic or ombrothrophic?).

Specific comments: - Please check if all abbreviations are explained when mentioned the first time - P3 mention trophic state of fen - P6 L20, …. …the higher the soil moisture, the more DOC is produced…. I doubt this very general statement. Permanent water logging reduces DOC production (anaerobic conditions). - P6 L 26…. What is DOC loss here, mineralization or export from the peatland or both? - P6 L32 …what is the DOC concentration in rain water? - P7 L 5 expand on how exactly the DOC model was calibrated - P7 L11 the period … was not simulated …. because exceptionally high rainfall……. Water coming from flooded rivers. I think it is probably a major weakness that the model cannot simulate heavy rain events because those are very important for DOC concentrations and release (connecting pools etc.) please comment on this how this affects the overall quality of the model. See also P8 L28 … decrease in model efficiency….. explained by exceptional events. Please explain why you think how the model is still useful if it cannot simulate rain/drought events and related preconditions. - Please avoid "good results", "satisfactory results" etc. as these are undefined terms or define what this means. - P10 L27 and Figure 5, what are upstream and downstream locations? This is discussed on page 10, but I could not find a description in the methods section. - Fig. 6 difference in DOC variation between control and rewetted site and discussion on page 9. I think this observation needs further discussion. I don't understand why DOC export (I would prefer "release" here)

is episodic at the rewetted site and I don't understand why this observation is consistent with the observation of Birkel et al., 2017 who used a stream discharge approach. Please explain! - Regarding a detailed explanation of the biogeochemical module, the authors refer to a previous paper. I think the authors should expand a little bit on this especially how draining and rewetting alters the quality of organic matter in comparison to the control site (density etc.). I also wonder why the authors believe that deeper peat is more aromatic. Please explain and include appropriate reference. - I suggest a language polishing by a native English speaker

---

## Referee Comment (RC2) · M Bechtold (Referee) · 6 Dec 2017

**Review of manuscript "Hydrological control of dissolved organic carbon dynamics in a rehabilitated Sphagnum–dominated peatland: a water-table based modelling approach" submitted to HESS by Bernard-Jannin et al.**

The paper deals with the effect of peatland rewetting/restoration on DOC dynamics. The study combines field monitoring (DOC pore water concentrations [DOC] and water table depth) with hydro-biogeochemical modeling. Two monitoring sites were installed within the same acidic fen peatland, one defines the control site (disturbed, drier conditions) and one the rewetted case due to recent (3 years) blocking of a ditch. Field data indicate a strong effect of the rewetting on the water table depth (>20 cm shallower water levels during summer) and higher [DOC] in summer in the rewetted location compared to the control. Also the quality of DOC seems to change due to rewetting. Modeling results indicate very contrasting fluxes at the two sites, fast runoff with very low evapotranspiration at rewetted sites vs. slow "deep" drainage at control site, which is interpreted to control DOC composition and export.

The basic scope of the paper (topic, field data, model approach) is interesting, relevant for the community and suitable for publication in HESS. The authors dealt with a challenging site because fluxes were not observed. Calibration of a hydrological model only on water table depth data with the objective to interpret resulting fluxes is questionable although maybe not impossible when setting sufficient constraints on calibration scheme and parameters, and when providing reliable uncertainty estimates. However, as the modeling, and in particular the calibration, has been conducted and presented now, one cannot trust the highly questionable simulated hydrological fluxes (e.g. extremely low ET at rewetted site). Fluxes are crucial for the [DOC] dynamics, e.g. ET causes a 'physical' accumulation of DOC in the pore water, thus one can also not rely on the following implications for DOC concentrations and exports. In my major points, I thus focus on the hydrological part of the paper. The DOC part is commented with less detail, although it must be also fundamentally revised because I assume that results will change considerably with an update of the hydrological modeling.

In general, as mentioned before, fluxes of a hydrological model calibrated only on water table depth should be interpreted carefully. I present my major concerns and give some additional recommendations afterwards.

**Major points**

I see several major problems in the hydrological modeling part of the study. The hydrological model is of fundamental importance for the results of the manuscript. The authors intent "to identify the main hydrological processes and factors controlling DOC dynamics" with their model which is based on a conceptual approach based on reservoirs. Modeled are two monitoring sites that lie in the same disturbed fen peatland. I assume the two sites were very similar before 2014 (same vegetation, 60 cm peat thickness, … same peat properties?). The only that changed in 2014 was the boundary condition for the rewetted site, which is effectively a higher drainage level due to ditch blocking.

**Separation into boundary-condition dependent and boundary-condition independent parameters:**

Having repeated the field situation, it is only possible to rely on the simulated fluxes when all model parameters that do not depend on changed boundary conditions (i.e. here the changed drainage level) are the same for both monitoring sites. As far as I understood the three

discharge coefficients (overland, lateral runoff through peat, and "deep" drainage) are dependent on the raised drainage level due to ditch blocking. As the conceptual model does not include a drainage level as a boundary condition, these parameters have to be tuned that simulated water levels match observed ones. *(This is actually a major weakness of the presented conceptual model, as for a more process-type of model only a single boundary condition would need to be changed and/or calibrated if not known precisely. Emphasize this disadvantage.).*
In contrast to the three discharge coefs, I consider the ET crop coefficients and infiltration rate from the peat-macro to the peat-matrix reservoir (Imax) as "boundary-condition-independent". They should be thus calibrated to the same value for both sites.

In the study, however, all these parameters were calibrated separately at both sites resulting in extreme differences for the crop coefficients and Imax at the two sites. As a consequence, e.g. ET is extremely low at the rewetted site. The authors recognized that this flux is unphysical and explain it by additional open water reservoirs (not modelled) that are laterally coupled to the modeled 1D profile and buffer water table depth dynamics during high water levels. I think this might be an important process and a valid interpretation, but a dramatic reduction of ET for the rewetted site is an unacceptable result for the scope of the paper. If this process (open water storage) is really relevant, it needs to be accounted for.

To further prevent such an "overfitting effect", I recommend that authors think about reliable constraints during the calibration. For example, I suggest to calibrate all parameters simultaneously for both sites keeping the "system boundary condition independent" parameters the same for both sites ( 'multi'-site calibration approach)

**Reliability of validation period:**

Validation with independent data is crucial. The authors used the third year of data for validation. This is basically ok. However, Nash-Sutcliffe efficiency (NS) drops from 0.8 (calib) to 0.3 (valid) for one of the sites. The enormous drop of performance is a strong indication of overfitting. Authors argue that it might be related to the exceptional flooding event between the second and the third year. If they are convinced that this is the reason, then the whole validation period is useless because we learn nothing about the reliability of the calibrated model from it. In fact, from what I can read and see in the paper, I have the impression that the third year can be used for validating the hydrology, because the initial water table depth of the validation period seems to be quite consistent between model and data. Or were the initial condition of the model set to the observed data? If yes this must be stated clearly, and the validation period should be shortened as validation NS would be overestimated by this manual adjustment of the state variable.

**Definition of reservoirs**

Sm is 0 at the lower peat boundary, why? Did authors determine in the lab that there are no macro-pores at 80 cm? At the surface the total porosity (Sm+Se) of the peat is defined to be 1. Justify also this decision, as it is not obvious how a material can have a porosity of 1. Did authors want to prevent a discontinuity between soil and surface storage? But why?

**DOC measurements**

The definition of the reservoirs is of particular importance also for the DOC interpretation. Were the sampling pipes emptied before taking samples? If not, it is not clear what the [DOC]

in the pipes actually is (mixture of recent rain water, [DOC] of soil pore water X days/weeks before sampling minux the losses due to being 'open water' in the pipe afterwards, etc.) . If yes, mainly the macro-pore water was refilling the empty pipe afterwards, i.e. authors need to compare the field data with modeled Sm [DOC], if I understood the approach correctly. Regarding the issue of sampling different soil pore spaces see also Zsolnay, 2003, Dissolved organic matter: artefacts, definitions, and functions, Geoderma.

**Effect of flooding event on DOC validation period**

What is the [DOC] in the river water? And do authors have any knowledge about how the flooding event affected the [DOC] of the peat profile in the third year? Authors need to discuss this. Till now, the third year is in particular questionable for validating the DOC model.

**Recommendations:**

Authors must establish much more confidence for their model. Besides taking into account my major and minor comments, I recommend that authors elaborate a sophisticated uncertainty analysis that provides uncertainty estimates for the flux components, i.e. that shows how variable the predicted flux components are for similarly performing models (similar fits of water table depth). Use probabilistic approaches like e.g. Bayesian modeling or GLUE. Authors can only generate sufficient confidence into the model results when authors are very critical with the calibration. For that, consider also the effect of optimizing the parameters for different objective functions, not just the daily NS or RMSE (e.g. different levels of aggregating the time series, daily, weekly, monthly averages) and add plausible errors to the forcing and water table depth data. These are ideas for directions authors need to look for.

Minor comments:

Page 4 - L15: "identify the locations" – I do not understand the sentence.

Page 5 – L4: What is Smax? Not defined after the equation but two paragraphs later.

Page 6 – L32: Why t+1 for PDOC and LDOC? Isn't it dependent on the concentration of time step t. Clarify this also in eqs 7 and 8

Page 7 – L14: How can the maximum porosity of the peat layer be equal to 1?

Page 7 – L14: "flooded river" … wording. The river itself cannot be flooded.

Page 7 – L20: Give both NS and RMSE for both the hydrological and the DOC model. Not clear why different skill metrics have been chosen for the two variables. Would be interesting to know NS of the DOC model. It seems to be barely better than just the mean of the observed concentrations (i.e. NS~0), in particular for the validation period.

Page 7 – L22: calibrated over the calibrated hydrol. model. …. improve wording.

Table 1: correct table, reservoirs and discharge description are confused !

Table 1: How can Imax as a rate have the unit [mm]? A rate is always per time.

Table 2: Format table correctly.

Michel Bechtold,
KU Leuven, BE; 6th Dec 2017.

---

## Author Comment (AC1) · 24 Jan 2018

The ms describes a modeling approach to predict the relationship between DOC production and hydrology, especially water table levels, in a fen which partly experienced draining and restoration. The authors used a parsimonious approach including a hydrological and a biogeochemical component. The model was calibrated directly at the site through monitoring of the peatlands water table level at the two measuring sites and by DOC analyses. In addition the authors carried out fluorescence measurements to determine changes in the DOM quality. In principal, several parsimonious models to predict the release of DOC dependent on hydrological changes from peatlands exist. Different from these models which commonly use discharge data to characterize the hydrological component, the authors use changes in water table to predict DOC release. The authors concluded that their model predicts DOC release at the two sites reasonably well (whatever this means). As an advantage compared to other models they argue that the use of water table changes instead of using discharge measurements allows predicting local changes in DOC release esp. if parts of a peatland underwent draining and restoration, which might have strong influence on DOC production and release. I don't have an emphasis on hydrological modeling of peatlands but on the biogeochemical processes of DOM production on peatlands, so I will not comment on the quality of the model itself.

I had a bit a hard time with the structure of the paper. One reason might be that the mentioned objectives of the study are very general . . . . to identify hydrological processes and factors controlling DOC dynamics . . . . . . the impact of rewetting on DOC export. It is known that water table changes is the major driver of DOC release in peatlands, because surface wetness determines the redox conditions on the mire's surface which in turn control microbial activity and peat decomposition. In this sense, it would be helpful if the authors could first describe in more detail what they expect the effect of peatland restoration on DOC release could be, especially what is expected how the amount and quality of DOM changes as result of drainage and subsequent rewetting. Based on this it would be easier to understand the chosen approach.

We will add a paragraph in the introduction explaining the expected results of restoration on DOC dynamics based on existing literature. Air Temperature and WTD are considered to be good descriptors for conceptual DOC models (Birkel et al., 2017; Lessels et al., 2015). Indeed these two parameters determine the microbial activity and peat decomposition that are the producing and consuming processes of DOC in peat water. Concerning studies on the impact of rewetting and drain blocking in peatland on DOC dynamics have shown opposite results with increasing concentrations (Hribljan et al., 2014; Strack et al., 2015) and decreasing concentrations (Höll et al., 2009; Wallage et

al., 2006). In this study we propose to use a modelling approach to simulate DOC concentrations in a rewetted peatland and identify factors of control that can explain differences in DOC dynamics.

In addition, we will specify the objectives of the study as follow: "The objectives of this study were to 1) identify the dominant hydrological processes in both the rewetted and undisturbed peatland locations, and 2) to understand how these hydrological processes affect the DOC dynamics in each of the two locations.

Moreover, I think that the comparison between the WTD and the stream discharge approach should be discussed in more detail. On page 12 the authors argue that the advantage of the WTD approach is that this model can predict DOC release at different areas in the same peatland in contrast to the stream discharge approach which integrates the entire peatland. However, the authors have only selected a single site with is under restoration and did not define which biogeochemical or hydrological component/factor is specific for the rewetted site. Due to this, it remains unclear how the specific behavior of the rewetted site compares to the overall variation of DOC release and DOM quality during water table changes in the entire peatland (compare the discussion in Birkel et al 2017 and Broder et al . 2015 cited therein).

The spatial variability of the biogeochemical or hydrological components is included in the range of the model parameters defined for the model calibration. Then the model is calibrated for both sites. It is therefore the values of the calibrated parameters that give an indication on the behavior (hydrology and DOC) of each site in order to explain the differences observed between both sites of the peatland.

This means that the observed changes at the single rewetted site provide a small data set only, so that it remains somehow speculative if the shown changes and effects are specific for rewetting sites. A larger data set which shows the general variability of WTD, DOM quality and DOC export at the selected peatland compared to the specific rewetted site would convince me that the chosen approach is of advantage compared to the one using the stream discharge approach.

Unfortunately we don't have the dataset as asked by the reviewer with in the same point WTD and DOC values. However it is clear that we have two locations with different water table and DOC dynamics, and that the model is able to explain these differences by differences in the partition between slow and rapid drainage. We will discuss this in the revised version of the ms by explaining that the different dynamics in the rewetted area might be explained by others factors than the restoration

The main reason of using a water table based model instead of a discharge based model is that it can be challenging to monitor discharge in peatlands (flat areas with many outlets varying in time and space). Using a water table based model allows to simulate hydrology when discharge data are missing as it is the case in our study site. Discharge based model would be more robust when simulating fluxes but water table based model has one advantage as it allows the application of the model in different locations within the same peatland. Therefore the use of a water table based model has to be seen as an alternative to the discharge based model when discharge data are not available.

I suggest that the authors discuss this in more detail including effects of spatial varying redox conditions (also hummocks and hollows), plant cover (what means Sphagnum dominated in this context), mineral redox barriers esp. iron-oxides; which are important for DOC dynamics in fens (is this site minerothrophic or ombrothrophic?).

In this study we mainly focus on the impact of hydrology on DOC dynamics therefore other factors such as redox are not discussed but their potential effects will be added in the new version of the ms. The plant cover is very homogeneous all over the site with a mixture of Graminoids (Mainly Molinia caerulea and in a much lower extent Eriophorum angustifolium and Ryhchospora alba), ericaceous shrubs (Erica tetralix and Calluna vulgaris) and sphagnum species. In addition, the site is an oligotrophic fen that has not developed any abundant hummock and hollow microtopography, such as in a typical ombrotrophic site

Specific comments:

- Please check if all abbreviations are explained when mentioned the first time

We'll check and correct after the final modifications.

- P3 mention trophic state of fen

It's oligotrophic. It will be added.

- P6 L20, . . . the higher the soil moisture, the more DOC is produced . . . . I doubt this very general statement. Permanent water logging reduces DOC production (anaerobic conditions).

We will remove this statement.

- P6 L 26 . . . What is DOC loss here, mineralization or export from the peatland or both?

Here, DOC loss corresponds to mineralization and sorption, it will be add in the ms.

- P6 L32 . . . what is the DOC concentration in rain water?

DOC concentration in the rain was measured in rain water samples and is 2 mg/L

- P7 L 5 expand on how exactly the DOC model was calibrated

The calibration of the model is explained in detail in the section 2.2.4

- P7 L11 the period . . . was not simulated . . . . because exceptionally high rainfall . . . . . . . Water coming from flooded rivers. I think it is probably a major weakness that the model cannot simulate heavy rain events because those are very important for DOC concentrations and release (connecting pools etc.) please comment on this how this affects the overall quality of the model.

The model is water balance model, it is therefore impossible to account for water coming from exceptional flood events. One have to keep in mind that we speak about very exceptional events (return period of about 50 years). Regular flood events can be simulated by the model. Moreover, the exceptional flood affected only Sr reservoir because peat profile was already fully saturated with

rain water when the flood occurred. Therefore the impact of the flood on peat water DOC should be very low. We will add this comment in the ms.

See also P8 L28 . . . decrease in model efficiency . . . . explained by exceptional events. Please explain why you think how the model is still useful if it cannot simulate rain/drought events and related preconditions.

This was indeed a weakness of the model. However, following the comments and suggestions of the 2$^{nd}$ reviewer we were able to improve the model calibration. And the model is now able to simulate with the same efficiency wet and dry condition (except exceptional rainfall events as discussed above)

- Please avoid "good results", "satisfactory results" etc. as these are undefined terms or define what this means.

We will define these expressions and assess the quality of the model based on the value of the efficiency criteria. In addition and following comments of the second reviewer, an uncertainty analysis have been performed to better assess the confidence we can have in the model.

- P10 L27 and Figure 5, what are upstream and downstream locations? This is discussed on page 10, but I could not find a description in the methods section.

Upstream is control and downstream is rewetted, we will make the changes.

- Fig. 6 difference in DOC variation between control and rewetted site and discussion on page 9. I think this observation needs further discussion. I don't understand why DOC export (I would prefer "release" here) is episodic at the rewetted site and I don't understand why this observation is consistent with the observation of Birkel et al., 2017 who used a stream discharge approach. Please explain!

We will change export to release. DOC release at the rewetted site is episodic because it is linked to the hydrology which is controlled by rapid drainage resulting from rain events. On the opposite DOC release in control site is linked to the hydrology which is controlled by slow drainage and is therefore less impacted by rain events. The observation is consistent with Birkel et al (2017) that found an episodic release of DOC (60% of release in 30% of the time). This is an analysis of the model results and is independent of the use of a water table or a discharge based model (both models simulate DOC release).

- Regarding a detailed explanation of the biogeochemical module, the authors refer to a previous paper. I think the authors should expand a little bit on this especially how draining and rewetting alters the quality of organic matter in comparison to the control site (density etc.). I also wonder why the authors believe that deeper peat is more aromatic. Please explain and include appropriate reference.

We will improve the explanation of the links between biogeochemical module and hydrology. We believe that DOM will be more aromatic in deep layer because it is "older" than in surface where the input of fresh organic matter would lead to less aromatic DOM as discussed in Strack et al (2015).

- I suggest a language polishing by a native English speake

We will send the paper to a corrector after the final changes are made.

References:

Birkel, C., Broder, T. and Biester, H.: Nonlinear and threshold-dominated runoff generation controls DOC export in a small peat catchment, J. Geophys. Res. Biogeosciences, 122(3), 498–513, doi:10.1002/2016JG003621, 2017.

Höll, B. S., Fiedler, S., Jungkunst, H. F., Kalbitz, K., Freibauer, A., Drösler, M. and Stahr, K.: Characteristics of dissolved organic matter following 20 years of peatland restoration, Sci. Total Environ., 408(1), 78–83, doi:10.1016/j.scitotenv.2009.08.046, 2009.

Hribljan, J. A., Kane, E. S., Pypker, T. G. and Chimner, R. A.: The effect of long-term water table manipulations on dissolved organic carbon dynamics in a poor fen peatland, J. Geophys. Res. Biogeosciences, 119(4), 577–595, doi:10.1002/2013JG002527, 2014.

Lessels, J. S., Tetzlaff, D., Carey, S. K., Smith, P. and Soulsby, C.: A coupled hydrology–biogeochemistry model to simulate dissolved organic carbon exports from a permafrost-influenced catchment, Hydrol. Process., 29(26), 5383–5396, doi:10.1002/hyp.10566, 2015.

Strack, M., Zuback, Y., McCarter, C. and Price, J.: Changes in dissolved organic carbon quality in soils and discharge 10years after peatland restoration, J. Hydrol., 527, 345–354, doi:10.1016/j.jhydrol.2015.04.061, 2015.

Wallage, Z. E., Holden, J. and McDonald, A. T.: Drain blocking: an effective treatment for reducing dissolved organic carbon loss and water discolouration in a drained peatland., Sci. Total Environ., 367(2-3), 811–21, doi:10.1016/j.scitotenv.2006.02.010, 2006.

---

## Author Comment (AC2) · 24 Jan 2018

**Answer to the review by Dr Bechtold**

We thank Dr Bechtold for the thoughtful review of the paper. We agree with all the points raised and we believe we can address them all in order to improve the quality of the paper. We explain below how we plan to address each of them in details.

**Review of manuscript "Hydrological control of dissolved organic carbon dynamics in a rehabilitated Sphagnum–dominated peatland: a water-table based modelling approach" submitted to HESS by Bernard-Jannin et al.**

The paper deals with the effect of peatland rewetting/restoration on DOC dynamics. The study combines field monitoring (DOC pore water concentrations [DOC] and water table depth) with hydro-biogeochemical modeling. Two monitoring sites were installed within the same acidic fen peatland, one defines the control site (disturbed, drier conditions) and one the rewetted case due to recent (3 years) blocking of a ditch. Field data indicate a strong effect of the rewetting on the water table depth (>20 cm shallower water levels during summer) and higher [DOC] in summer in the rewetted location compared to the control. Also the quality of DOC seems to change due to rewetting. Modeling results indicate very contrasting fluxes at the two sites, fast runoff with very low evapotranspiration at rewetted sites vs. slow "deep" drainage at control site, which is interpreted to control DOC composition and export.

The basic scope of the paper (topic, field data, model approach) is interesting, relevant for the community and suitable for publication in HESS. The authors dealt with a challenging site because fluxes were not observed. Calibration of a hydrological model only on water table depth data with the objective to interpret resulting fluxes is questionable although maybe not impossible when setting sufficient constraints on calibration scheme and parameters, and when providing reliable uncertainty estimates. However, as the modeling, and in particular the calibration, has been conducted and presented now, one cannot trust the highly questionable simulated hydrological fluxes (e.g. extremely low ET at rewetted site). Fluxes are crucial for the [DOC] dynamics, e.g. ET causes a 'physical' accumulation of DOC in the pore water, thus one can also not rely on the following implications for DOC concentrations and exports. In my major points, I thus focus on the hydrological part of the paper. The DOC part is commented with less detail, although it must be also fundamentally revised because I assume that results will change considerably with an update of the hydrological modeling.

In general, as mentioned before, fluxes of a hydrological model calibrated only on water table depth should be interpreted carefully. I present my major concerns and give some additional recommendations afterwards.

**Major points**

I see several major problems in the hydrological modeling part of the study. The hydrological model is of fundamental importance for the results of the manuscript. The authors intent "to identify the main hydrological processes and factors controlling DOC dynamics" with their model which is based on a conceptual approach based on reservoirs. Modeled are two monitoring sites that lie in the same disturbed fen peatland. I assume the two sites were very similar before 2014 (same vegetation, 60

cm peat thickness, … same peat properties?). The only that changed in 2014 was the boundary condition for the rewetted site, which is effectively a higher drainage level due to ditch blocking.

***Separation into boundary-condition dependent and boundary-condition independent parameters:***

Having repeated the field situation, it is only possible to rely on the simulated fluxes when all model parameters that do not depend on changed boundary conditions (i.e. here the changed drainage level) are the same for both monitoring sites. As far as I understood the three discharge coefficients (overland, lateral runoff through peat, and "deep" drainage) are dependent on the raised drainage level due to ditch blocking. As the conceptual model does not include a drainage level as a boundary condition, these parameters have to be tuned that simulated water levels match observed ones. (*This is actually a major weakness of the presented conceptual model, as for a more process-type of model only a single boundary condition would need to be changed and/or calibrated if not known precisely. Emphasize this disadvantage*.)

It is true that only one boundary condition would need to be changed for a process-type model, but additional(s) parameter(s) would be required to describe the process. Process-type models at the interface between surface and peat waters, including exchange between these two compartments could be complex and difficult to calibrate. The idea to use a more conceptual model is to have a model with few input parameters needed (at the cost of increase number of calibrated parameter indeed). This disadvantage of conceptual model over process based model will be stated in the ms.

In contrast to the three discharge coefs, I consider the ET crop coefficients and infiltration rate from the peat-macro to the peat-matrix reservoir (Imax) as "boundary-condition independent". They should be thus calibrated to the same value for both sites. In the study, however, all these parameters were calibrated separately at both sites resulting in extreme differences for the crop coefficients and Imax at the two sites. As a consequence, e.g. ET is extremely low at the rewetted site. The authors recognized that this flux is unphysical and explain it by additional open water reservoirs (not modelled) that are laterally coupled to the modeled 1D profile and buffer water table depth dynamics during high water levels. I think this might be an important process and a valid interpretation, but a dramatic reduction of ET for the rewetted site is an unacceptable result for the scope of the paper. If this process (open water storage) is really relevant, it needs to be accounted for. To further prevent such an "overfitting effect", I recommend that authors think about reliable constraints during the calibration. For example, I suggest to calibrate all parameters simultaneously for both sites keeping the "system boundary condition independent" parameters the same for both sites ( 'multi'-site calibration approach).

We agree with the comments of Dr Bechtold and redid the calibration by keeping ET and Imax coefficients similar for both sites in order to perform a multi-site calibration. In addition we decided to improve the choice of the calibration period by using 2014 and 2017 that are the driest and wettest years. 2015 is now used as a validation. In addition, we present additional efficiency criteria to assess the quality of the model (br² and RMSE). The new calibrated parameters are presented below (table 1, Fig. 1). We constrained the ET coefficients values to be in a range in agreement with observed values (Lafleur et al., 2005). The objective function is now the sum of NS for the two locations (control and rewetted) and two validation periods (2014 and 2017). We can see from the results that, despite being less performing than in the first version of the paper, the model is still able to reproduce water table for both site when taking into account a multi site calibration approach,

with little differences in the coefficients found in the first version of the ms (except for evapotranspiration downstream). NS coefficients are all positives; br2 all greater than 0.2 and RMSE are between 1 and 9 cm. The model performed better for the control site in the dry year than in the wet year and the opposite can be observed for the rewetted site. Finally, we don't observe a drop of performance of the model for the validation period.

Table 1: Updated calibrated parameters and performance criteria based on a multi-site calibration approach. Range of parameters calibration are also indicated)

|  | REWETTED | CONTROL | Range |
|---|---|---|---|
| Kcd | 0.37 | | 0.01 – 0.5 |
| Kcg | 0.40 | | 0.4 – 0.8 |
| Imax | 0.84 | | 0.2 -5 |
| αp | 1.6E-05 | 1.9E-04 | 0 -0.01 |
| αr | 0.20 | 0.37 | 0 -0.5 |
| αsr | 0.20 | 0.27 | 0 -0.5 |
| NS calib wet (2014) | 0.1 | 0.61 | |
| br2 calib wet (2014) | 0.52 | 0.67 | |
| RMSE calib wet (2014) | 0.01 | 0.01 | |
| NS calib wet (2017) | 0.25 | 0.16 | |
| br2 calib wet (2017) | 0.26 | 0.24 | |
| RMSE calib wet (2017) | 0.065 | 0.080 | |
| NS valid (2015) | 0.10 | 0.30 | |
| Br2 valid (2015) | 0.54 | 0.39 | |
| RMSE valid (2015) | 0.02 | 0.09 | |

[Figure]

Figure 1: (a) Time series of meteorological data (PET, potential evapotranspiration and P, precipitation) used as input data in the hydrological model, (b) simulated and observed WTD in the rewetted site and (c) simulated and observed WTD in the control site. Calibration and validation periods are also indicated.

**Reliability of validation period:**

Validation with independent data is crucial. The authors used the third year of data for validation. This is basically ok. However, Nash-Sutcliffe efficiency (NS) drops from 0.8 (calib) to 0.3 (valid) for one of the sites. The enormous drop of performance is a strong indication of overfitting. Authors argue that it might be related to the exceptional flooding event between the second and the third year. If they are convinced that this is the reason, then the whole validation period is useless because we learn nothing about the reliability of the calibrated model from it. In fact, from what I can read and see in the paper, I have the impression that the third year can be used for validating the hydrology, because the initial water table depth of the validation period seems to be quite consistent between model and data. Or were the initial condition of the model set to the observed data? If yes this must be stated clearly, and the validation period should be shortened as validation NS would be overestimated by this manual adjustment of the state variable.

As stated in the previous point, we modify calibration and validation period. Calibration is performed on the driest and the wettest year (2014 and 2017) and validation is perfomed on an intermediate year (2016). For the simulation of 2017, the starting H is set as the observation, this will be made clear in the ms. With the new calibration strategy (multi sites) and the change of calibration/validation periods we don't observe a drop of performance between calibration and validation.

**Definition of reservoirs**

Sm is 0 at the lower peat boundary, why? Did authors determine in the lab that there are no macro-pores at 80 cm? At the surface the total porosity (Sm+Se) of the peat is defined to be 1. Justify also this decision, as it is not obvious how a material can have a porosity of 1. Did authors want to prevent a discontinuity between soil and surface storage? But why

The conceptual model was build in a way to obtain a good compromise between the number of parameters needed (2 parameters, Hmax and Θinf) and a solid description of the flows at the interface between soil and surface. Sm reservoir was conceptualized to be a transitional reservoir between 100% Se peat at the lower boundary and a 100% Sr surface at the upper boundary. We thus made the following assumptions: macroporosity is null at the bottom of the peat layer and the total porosity is one at the surface, which is in agreement with observed values (Bourgault et al., 2017).

**DOC measurements**

The definition of the reservoirs is of particular importance also for the DOC interpretation. Were the sampling pipes emptied before taking samples? If not, it is not clear what the [DOC] in the pipes actually is (mixture of recent rain water, [DOC] of soil pore water X days/weeks before sampling minux the losses due to being 'open water' in the pipe afterwards, etc.) . If yes, mainly the macro-pore water was refilling the empty pipe afterwards, i.e. authors need to compare the field data with modeled Sm [DOC], if I understood the approach correctly. Regarding the issue of sampling different soil pore spaces see also Zsolnay, 2003, Dissolved organic matter: artefacts, definitions, and functions, Geoderma.

Indeed, the sampling pipes were emptied before sampling, so we can assume that we measure mainly [DOC] in Sm. We propose to calibrate production and loss constant on [DOC] in Sm. New

results are shown in Fig 2. The model still perform well for after the changes except fot the year 2017 in control were concentration are systematically over estimated. This can be related to the difficulties of the model to simulate water table for this period. However the model simulated lower concentrations in rewetted than in control as observed. Calibrated parameters (only 2 parameters now as only DOC in Sm is simulated) and efficiency criteria are shown in table 2. Despite a relatively high RMSE, the model is able to represent the trend for most of the simulated periods. This is in agreement with the fact that hydrology plays a key role in controlling [DOC] dynamics through the water table depth as stated in the first version of the ms.

[Figure]

Figure 2: Simulated [DOC] in Sm and observed [DOC] after new calibration

Table 2: Calibrated parameter and efficiency criteria of the DOC model

|  |  | Rewetted | Control |
|---|---|---|---|
|  | Kprod Sm | 5.00E-08 | 9.00E-07 |
|  | Kloss Sm | 5.00E-04 | 1.20E-02 |
| Calib (2014) | RMSE (mg.L$^{-1}$) | 5.4 | 1.6 |
|  | Br2 | 0.003 | 0.89 |
| Calib (2017) | RMSE (mg.L$^{-1}$) | 8.6 | 8 |
|  | Br2 | 0.18 | 0.03 |
| Valid (2015) | RMSE (mg.L$^{-1}$) | 8.9 | 10.8 |
|  | Br2 | 0.34 | 0.31 |

**Effect of flooding event on DOC validation period**

What is the [DOC] in the river water? And do authors have any knowledge about how the flooding event affected the [DOC] of the peat profile in the third year? Authors need to discuss this. Till now, the third year is in particular questionable for validating the DOC model.

The DOC from the river during the flood was measured and is equal to 12 mg.L$^{-1}$. We think that this 50 years return period flood affected only Sr reservoir because the peat profile was already fully saturated with rain water when the flood reached the peatland. Therefore the impact of the flood on peat water and on DOC is expected to be negligible.

**Recommendations:**

Authors must establish much more confidence for their model. Besides taking into account my major and minor comments, I recommend that authors elaborate a sophisticated uncertainty analysis that

provides uncertainty estimates for the flux components, i.e. that shows how variable the predicted flux components are for similarly performing models (similar fits of water table depth). Use probabilistic approaches like e.g. Bayesian modeling or GLUE. Authors can only generate sufficient confidence into the model results when authors are very critical with the calibration. For that, consider also the effect of optimizing the parameters for different objective functions, not just the daily NS or RMSE (e.g. different levels of aggregating the time series, daily, weekly, monthly averages) and add plausible errors to the forcing and water table depth data. These are ideas for directions authors need to look for.

Following the recommendations of Dr Bechtold, we performed an uncertainty analysis to better assess the confidence of the simulated fluxes. We ran a GLUE analysis with 300 000 runs for each location and using a criteria of NS>0.2 to select behavioral simulations. The ranges of each parameters used in the analysis are the same than in table 1.10th and 90th percentiles of each fluxes are presented in table 3.

Table 3: 10th and 90th percentiles of the fluxes resulting for the GLUE analysis (300 000 runs, behavorial runs for NS>0.2)

| | Rewetted | | Control | |
|---|---|---|---|---|
| | 10th | 90th | 10th | 90th |
| ET (mm.yr$^{-1}$) | 307 | 384 | 349 | 468 |
| R (mm.yr$^{-1}$) | 94 | 294 | 24 | 242 |
| D (mm.yr$^{-1}$) | 2 | 33 | 37 | 349 |
| O (mm.yr$^{-1}$) | 199 | 390 | 0 | 190 |

Uncertainty on fluxes is high (as we can expect from the use of a conceptual model) but we can see significant differences concerning the partition between drainage and overland flow for the two locations (slow deep drainage dominated upstream and rapid surface drainage dominated downstream). These findings are in agreement with results presented in the first version of the ms. We plan to include the results of the uncertainty analysis (included a detailed description in the method) in the paper to strengthen our conclusions.

All the changes on the hydrological model calibration have been transfer to the DOC model as discussed above. The DOC balance has been recalculated including uncertainty analysis results (table 4). Results and discussion parts will be modified according to the new findings.

Table 4: DOC release for control and rewetted location over the studied period. Best indicates values for the best set of parameters. 10th and 90th percentiles resulting of the uncertainty analysis are also indicated.

| | rewetted | | | control | | |
|---|---|---|---|---|---|---|
| | Best | 10th | 90th | Best | 10th | 90th |
| DOC release (gC m$^{-2}$ yr$^{-1}$) | 1.9 | 1.6 | 5.6 | 2.4 | 0.9 | 9.0 |

*Minor comments:*

Page 4 - L15: "identify the locations" – I do not understand the sentence.

It will be rephrase: "to identify the significant differences between the factors"

Page 5 – L4: What is Smax? Not defined after the equation but two paragraphs later.

Smax will be defined just after the equation.

Page 6 – L32: Why t+1 for PDOC and LDOC? Isn't it dependent on the concentration of time step t. Clarify this also in eqs 7 and 8

This was a mistake in writing the equations. Produced and consumed DOC are indeed dependent on time step t.

Page 7 – L14: How can the maximum porosity of the peat layer be equal to 1?

The maximum porosity represents the porosity at the surface, which is one.

Page 7 – L14: "flooded river" … wording. The river itself cannot be flooded

It will be change to : "the water coming from the river during floods"

 Page 7 – L20: Give both NS and RMSE for both the hydrological and the DOC model. Not clear why different skill metrics have been chosen for the two variables. Would be interesting to know NS of the DOC model. It seems to be barely better than just the mean of the observed concentrations (i.e. NS~0), in particular for the validation period.

The reason we didn't add the NS for the DOC is that NS is especially relevant when considering continuous time series with high variation (peaks in discharge or water table data). As we don't have many DOC measurements and the variations is not as high as for water table data, we think that NS is not relevant to correctly describe the efficiency of the DOC model and that is why we just presented the RMSE that give quantitative information on the model performance. It is not unusual to not present NS when quality data are scarce, e.g. Garneau et al. (2017). We propose to present NS, Br2 and RMSE for the hydrology and RMSE and Br2 for the DOC model.

Page 7 – L22: calibrated over the calibrated hydrol. model. …. improve wording.

We will change to:"the DOC module was calibrated after the calibration of the hydrological model"

Table 1: correct table, reservoirs and discharge description are confused !

We will improve the description of discharge and reservoirs.

 Table 1: How can Imax as a rate have the unit [mm]? A rate is always per time. Table 2: Format table correctly. Michel Bechtold, KU Leuven, BE; 6th Dec 2017.

This will be change. Imax is defined in mm in the original paper. The time unit was implicit and corresponds to the model time step. It will be change to mm.day$^{-1}$ (as the model uses a daily time step).

References:

Bourgault, M.-A., Larocque, M. and Garneau, M.: Quantification of peatland water storage capacity

using the water table fluctuation method, Hydrol. Process., doi:10.1002/hyp.11116, 2017.

Garneau, C., Sauvage, S., Sánchez-Pérez, J. M., Lofts, S., Brito, D., Neves, R. and Probst, A.: Modelling trace metal transfer in large rivers under dynamic hydrology: A coupled hydrodynamic and chemical equilibrium model, Environ. Model. Softw., 89, 77–96, doi:10.1016/j.envsoft.2016.11.018, 2017.

Lafleur, P. M., Hember, R. A., Admiral, S. W. and Roulet, N. T.: Annual and seasonal variability in evapotranspiration and water table at a shrub-covered bog in southern Ontario, Canada, Hydrol. Process., 19(18), 3533–3550, doi:10.1002/hyp.5842, 2005.

---

## Author Response (AR2)

Dear Editor,

First of all, thank you for having taken the time to review our paper. We could address all your comment and our answers are below.

* Discussion of the conceptual model: In the introduction (lines 21-29 on page 2 in the revised manuscript) it is argued why a conceptual model is used. After the discussion with the second reviewer, it was written in the reply that the model would in the revised manuscript be discussed more, including also the drawbacks. I think this is still missing (but I might have missed it, in this case the comment is not valid). I read the manuscript before going through the discussion and I stumbled over this point as well (same as the reviewer). While the argumentation about the number of parameters is reasonable and it seems reasonable to use such a model for the purpose of the paper, there are not only advantages of a conceptual model. The dependence of parameters on boundary conditions is already mentioned by the reviewer. However, also predictions made with a conceptual model that go out of the range of the data that were used for calibration and validation have to be seen critically. As the model parameters are not physically based parameters that are related to material properties only, one has to be careful with making predictions. A physically based model is more reliable in this respect. At least one sentence discussing modeling approaches in a more balanced way would be appropriate.

In order to balance the benefits of conceptual vs physically based model we added the following sentence. "Nevertheless, these parameters have to be adjusted to every condition through calibration and validation phases when a more physically model would only require adjusting boundary conditions. In addition, conceptual models are valid for a specific range of input data and should not be used for prediction where conditions lie out of their validation range."

* Section 2.2.2 (DOC model): I am confused with the mass balance. What are the units of DOC in eq. (9)? Intuitively, they should be mass / volume. However, if the balance is right, DOC has to be a flux (unit mass / (length * time)). It is not clear what flux this should be. Also: That would mean there is no storage in the model, just fluxes that add to zero. Also, there is no time step involved (Delta t = $t^{(n+1)} - t^{(n)}$). This does not make sense to me. The right hand side of eq. (9) is all taken at the old time step t (if I get it right). That would mean I could calculate the flux DOC (if it is a flux) at any time t+1 and would get the same result, no matter what I choose for t+1. It would not even depend on the time span between t and t+1.
Usually a balance with explicit time integration reads
($Mass^{(t+1)} = Mass^{(t)} + (Fluxes\ in - Fluxes\ out)^{(t)} * Area * Delta\ t$.
I recommend to check the balance and if it is correct to explain why DOC is here a flux and why there is no storage and why the flux at the new time does not depend on the time span between old and new time.

We acknowledge that the units were confusing. the mass balance and units were checked and corrected. $DOC^t_S$ are the mass auqntity of DOC in eqch reservoir (storage). Fluxes in are PDOC and infiltrated DOC (from rain or Sm , depending on the considered reservoir), fluxes out are CDOC and DOC Infiltration or drainage. The time step was implicit since it is constant and daily (delta t=1), we have had it for clarity. The model is a 1D vertical model and the results can be expressed per unit area (here per m²). Finally we have an equation following $Mass^{(t+1)} = Mass^{(t)} + (Fluxes\ in - Fluxes\ out)^{(t)} * Delta\ t$ with all terms expressed per unit area and all units are consistent.

*Line 15 on page 7 : Should be 4 parameters, and not 6, right?

Yes, correction made.

*Section 2.2.4, lines 2-3 on page 8: What was the Nash-Sutcliff coefficient calculated for? The water table depth? Please specify. Also: How many observations were used?

NS is calculated for water table depth, it was added in the text. The observation period for calibration and validation are described in 2.2.3

*Line 26 on page 8: I would not only refer to Figure 3, but also to Figures 4 and 5 as you mention here both WTD and DOC.

Yes, the text was modified accordingly

* Line 4, page 9: What means p<0.001? Please specify. Same page: What is a PARAFAC analysis

p is the p value of the Tukey test (2.1.2), we changed to p-value<0.001

PARAFAC is the parallel factor analysis (2.1.2). "PARAFAC is a commonly used method to analyze EEMs based on the decomposition of DOM fluorescence signature into individual components that provide estimates of the relative contribution of each component to total DOM fluorescence (Fellman et al., 2010). " was added in 2.1.2

* Line 1, page 10: How is 'significant' quantified? If not at all, I would suggest to not use the word.

We considered the difference significant since the 2 flux ranges derived from the uncertainty analysis do not overlap 188-394 vs 0-40mm for overflow and 1-79 vs 102-144mm for drainage. We added our description of significant in text. "the difference between overland flow and drainage flow in the two sites can be considered significant as the ranges provided by the uncertainty analysis do not overlap"

* Lines 6-7, page 10: I was puzzled. From Fig. 5 one gets the impression that the triangles lie all above the model line and the circles too for the first period. This would be a systematic underestimation or not?

Not all the triangle and circles lie over the model line although a majority does. In addition if we consider the uncertainty in the measurement, we believe there is no systematic overestimation in this case.

* Section 4.1. on page 11: I agree that a model helps to understand the characterization of fluxes in the different systems. Nevertheless: It is still a model with no comparison or control of measurements of fluxes. One has to keep in mind that it is still an interpretation and not a 'shown truth'.
Yes, although in this case we believe that the uncertainty analysis can help in assessing the validity of the model Even though, we agree that the model is not a "shown truth". A sentence was added 4.2.1 : "However, these results can only be considered an interpretation as there are no measurements of fluxes"